# Post-earthquake rapid resealing of bedrock flow-paths by concretion-forming resin
Hidekazu Yoshida [1] ✉, Koshi Yamamoto[1], Yoshihiro Asahara [2], Ippei Maruyama[2,3], Koichi Karukaya[1,4], Akane Saito[4], Hiroya Matsui[5], Akihito Mochizuki[5], Mayumi Jo[6], Nagayoshi Katsuta[7], Ayako Umemura[1] & Richard Metcalfe [8]

Many underground activities may require reducing or preventing fluid flows through bedrock, e.g., sealing of site investigation boreholes, underground tunneling, hydrocarbon field abandonment, and nuclear waste disposal. Cementitious materials such as grout are commonly used for bedrock flow-path sealing, however conventionally used these materials unavoidably undergo physical and chemical degradation, therefore potentially decreasing seal durability. Here, we report a more durable sealing method for concretion-forming resin developed by learning from natural calcite, $CaCO_3$, and spheroidal concretion formation. The method was tested by sealing flow paths next to a tunnel in an underground research laboratory at 350 m depth, in Hokkaido, Japan. The flow paths were initially sealed rapidly, then resealed after disturbance by repeated earthquakes with foci below the underground research laboratory at depths of 2–7 km and maximum magnitude $Mw$ 5.4. The treated rock mass rapidly recovered its very low natural permeability, demonstrating robust self-sealing and healing.

Effectively permanent sealing of potential bedrock flow paths, such as boreholes or underground excavations, and damaged rock around such features, is often required by international guidance and other stakeholders to build confidence in the safety and effectiveness of many types of underground activity[1–3]. Here "sealing" should not be understood to necessarily mean complete prevention of fluid flow, but rather reduction of fluid fluxes to some acceptable level that is dependent on the context, such as fluid fluxes no greater than natural fluxes through the rock mass.

Examples of sealing applications include abandonment of wells such as those used for hydrocarbon exploration or exploitation, sealing of boreholes used for site characterization of underground waste repositories, and potential sealing of excavation damaged zones (EDZs) around waste galleries and waste pits[1–8]. Since many of these activities are concerned with the isolation and containment of materials in deep bedrock effectively permanently (at least many thousands to hundreds of thousands of years), where seals are needed (e.g., in boreholes, in damaged zones around excavations) they will need to be effective indefinitely. Most commonly, flow paths are sealed using cementitious materials such as ordinary Portland cement and low-pH cement. However the durability of these materials may be insufficient to achieve permanent sealing due to the leaching of key constituents (e.g., $Ca(OH)_2$) by formation waters e.g., refs. 9,10, or chemical alteration, such as the formation of expansive phases like ettringite leading to cracking[11–13]. The durability of such conventional grouting represented by cementitious materials (e.g., cement-milk) is empirically understood for certain periods of time[14]. However, the sealing effect commonly decreases over time, especially in cases where flowing water contacts the cement, and therefore in some applications, additional grouting may be necessary to maintain the effectiveness of the seals[15–17]. Cementitious seals may be effective in usual civil engineering applications (which consider short timescales of perhaps a few decades), or in cases where water flows are very small (such as in boreholes/wells drilled in many mudrocks). However, cementitious seals may not be sufficiently durable where very long-term sealing performance is required.

In addition, in many applications, conventional cement-based seals are emplaced by injection at rather high pressures that exceed the in situ pore-water pressure[18,19]. Such injection can cause new fractures to form, and/or enhance the connectivity of existing flow-paths, thereby increasing the hydraulic conductivity of the rock and decreasing the long-term barrier

[1]Nagoya University Museum, Nagoya University, Nagoya, Japan. [2]Graduate School of Environmental Studies, Nagoya University, Nagoya, Japan. [3]School of Engineering, The University of Tokyo, Tokyo, Japan. [4]Sekisui Chemical Co., LTD, Ritto, Japan. [5]Japan Atomic Energy Agency, Horonobe, Hokkaido, Japan. [6]Taisei Corporation, Shinjuku-ku, Tokyo, Japan. [7]Faculty of Education, Gifu University, Gifu, Japan. [8]Quintessa Limited, Henley-on-Thames, Oxfordshire, UK.
✉ e-mail: dora@num.nagoya-u.ac.jp

**Fig. 1 | Concretion formation process determined by studying Tusk-shell concretions[28,29]. a** Tusk-shell (*Fissidentalium* spp.) concretion observed in the Yatsuo Formation of Toyama Prefecture, Japan. **b** Model of concretion formation and an equation that can be used to estimate the formation rate of a concretion. Conceptual view of the features of spheroidal concretions. Ca profile and porosity distribution across the spherical concretion formed under rather stable conditions under which solutes diffused continuously, causing calcite precipitation until the organic source of carbon at the center of the concretion was consumed[26]. Scale bar = 1 cm.

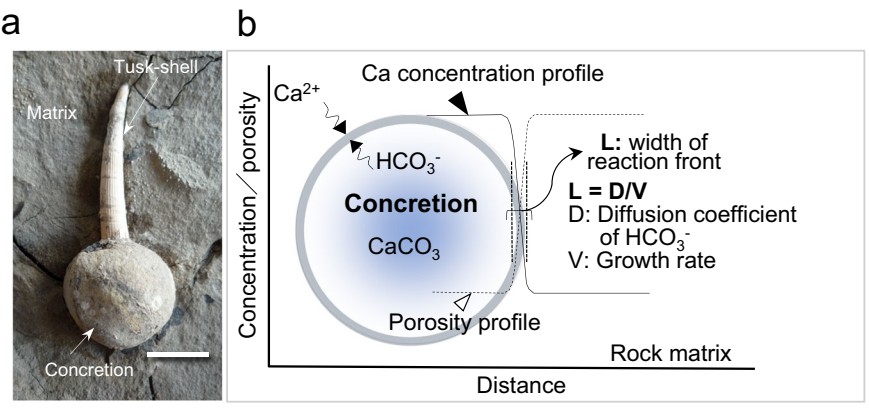

function of the geosphere[20]. Furthermore, physical perturbations of seals e.g., those caused by earthquakes, must be also taken into account for any kind of long-term usage of the underground environment.

Bentonite (a smectite-dominated material) is an alternative material that has been proposed to seal some radioactive waste repositories because if emplaced in a dry state, either as bentonite blocks or as a mixture with sand or crushed rock, it swells upon hydration and develops swelling pressure dependent on the dry density[21]. This swelling may reduce the apertures, and hence connectivities, of fractures in the EDZ or natural fractures, thereby helping to seal the EDZ. This is important for radioactive waste repositories constructed in certain types of deep bedrock (notably fractured crystalline rock), in which a network of flow paths will inevitably develop in an EDZ around the underground excavations[22–24]. It is crucial to prevent the EDZ from forming a pathway for the unacceptably rapid transport of radioactive and other contaminants[24] and to prevent water fluxes sufficiently rapid to be detrimental to the engineered barrier system. However, bentonite may be unsuitable where it is necessary to seal more transmissive flow paths because in the long term, it may be eroded by flowing groundwater and/or may be chemically incompatible with the prevailing groundwater chemistry[21,25].

To address the problem of existing materials having insufficient durability under certain circumstances, by learning from the natural formation process of spheroidal calcium carbonate concretions, we have successfully developed a rapid and long-lasting sealing technology. This technology stimulates calcite ($CaCO_3$) precipitation along flow paths or in matrix pores by using a 'concretion-forming resin'. The aim is to reproduce the process by which spheroidal calcite concretions form naturally.

Spheroidal calcium-carbonate concretions are frequently observed in marine sediments of widely varying geological ages and often contain well-preserved fossils[26–28]. Based on examining several hundred natural concretions we have found that they can grow in poorly consolidated sediment within several years and may even reach a size of meters[29,30] (Fig.1a). Detailed studies of concretions have shown that the formation of a spheroidal concretion can be explained by diffusion of $HCO_3^-$ outwards from a decomposing organism inside, leading to a syn-depositional reaction front where the $HCO_3^-$ combines with $Ca^{2+}$ in the surrounding porewater, causing supersaturation of $CaCO_3$ (calcite) which precipitates causing outward concretion growth[28–30] (Fig. 1b). During concretion growth, calcite fills pore spaces even down to sizes <1 μm[25], dramatically reducing the porosity and permeability of the sediments[31] (Fig.1b). Due to this, fossils inside concretions are often well preserved because water penetration is restricted and dissolution is limited. The sealing of pore space can mechanically harden the sediments[32] and the hydraulic conductivity of the concretion reduces by at least two to three orders of magnitude compared to that of the surrounding sedimentary matrix[31,32]. This reduction in permeability eventually provides strong resistance to degradation processes such as weathering for millions of years[31].

The 'concretion-forming resin' we have developed reproduces the rapid calcite concretion formation process seen in nature thereby resisting groundwater penetration. The resin (commercial product name: CRJ; Supplementary Note 1) is formed from two agents mixed in a ratio 1:1. One of these agents is 'bisphenol A epoxy resin' and 'glycidyl ether of poly-hydroxyalkan' containing C, H, and O oligomer. With this is mixed a hardening agent consisting of a mixture of 'reactant of isocyanate and organic amine' and 'monoalkylene amine' containing C, H, N, and O[33]. After hardening the solubility of the resin is quite low (order of ppb at 25 °C)[34] and the permeability is also low (<$10^{-10}$ m/s)[35]. However, the resin is able to hydrate and any molecule that includes $H_2O$ can diffuse through it if a concentration gradient exists[36,37]. This property is maintained within a temperature range of −50 °C to 200 °C, under any redox condition and at pH > 4, although it is most effective under neutral to alkaline conditions[38,39].

The 'concretion-forming resin' contains ions needed to facilitate the formation of a concretion, i.e., it causes calcite precipitation within fractures and pores around the resin-impregnated rock. The concentration of ions within the resin can also be varied to promote concretion formation at the required rate, taking into account the characteristics of flow paths in the host rock and groundwater chemical conditions. Also taken into account in formulating the resin is the changing release rate of ions (e.g., Ca) due to clogging of flow paths. The 'concretion-forming resin' itself increases in hardness after a few tens of minutes, and after hardening it continuously supplies the source ions for concretion formation until these ions are consumed. The hardening speed and the duration of ion supply for concretion formation (calcite precipitation) after hardening can be controlled by adjusting the resin formulation. For example, the 'concretion-forming resin' can be in the form of liquid and/or pellets, or micron-sized capsules (Supplementary Note 1).

In the tests described below, the injected liquid type of 'concretion-forming resin' was used and initially hardened in the bedrock. After hardening the resin causes porewater to super-saturate with respect to calcite by releasing high concentrations of $HCO_3^-$ and $Ca^{2+}$ that migrate into fracture flow-paths and micro-pores of adjacent rock matrices. Eventually, a calcite seal inevitably develops in and around the resin. The concretion-forming ions diffuse outwards from the resin due to a concentration gradient that develops even under high porewater pressure conditions in deep bedrocks. By this process, flow paths and/or pores will be sealed by calcite precipitation after the hardening of the resin. This sealing will occur without any mechanical disturbance within any matrix pores or fractures that have diameters/apertures in the order of microns or less. Furthermore, if the bedrock groundwater, whether fresh or saline, contains $HCO_3^-$ and $Ca^{2+}$, it will even more rapidly reach super-saturation with calcite, increasing further the rate at which pore space is filled and open fractures that act as flow paths are sealed. However, further research is required to establish the precise range of water salinities and compositions over which the resin will be effective. New calcite overgrowths on pre-formed calcite crystals can also be

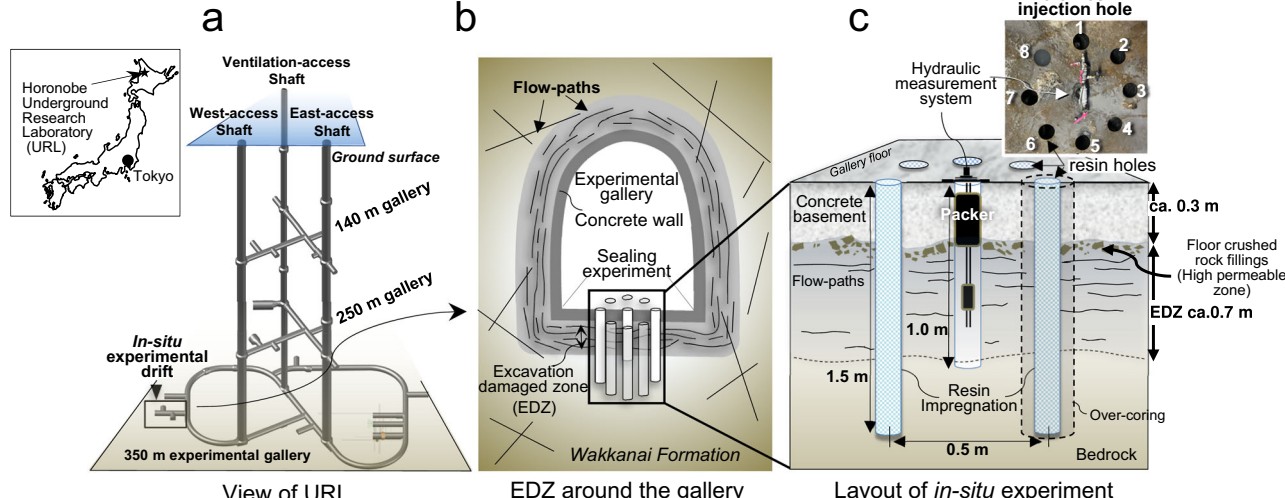

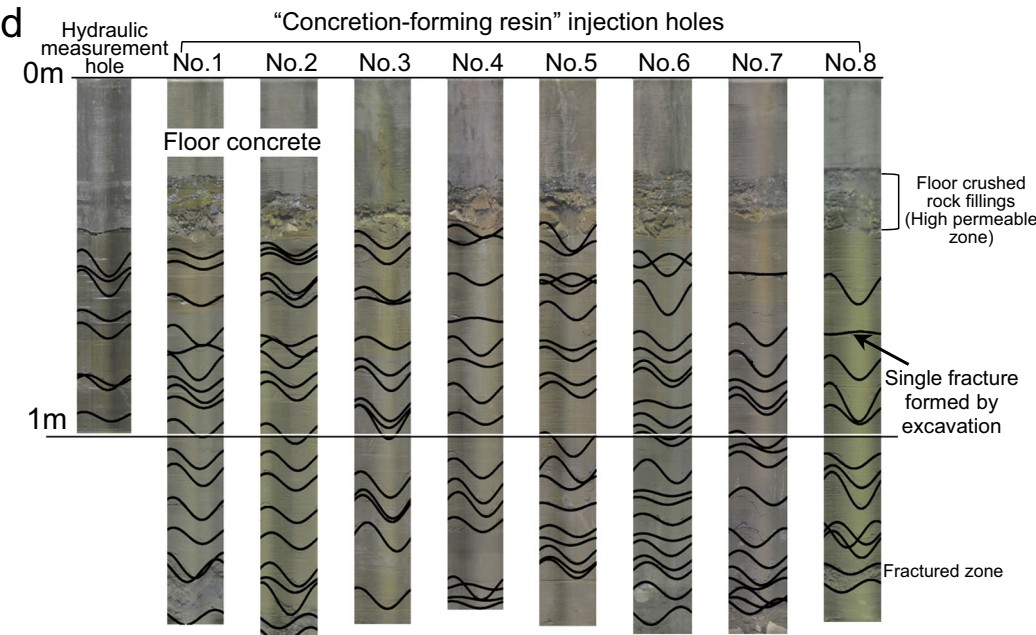

EDZ developed in the floor bedrock has a high number of fracture
flow-paths formed due to excavation.

**Fig. 2 | Conceptual view of URL, EDZ, and in situ experiment. a** Horonobe URL excavated in the Wakkanai formation down to 350 m from the ground surface. **b** Excavated damaged zone (EDZ) developed around the gallery identified as 'URL experimental drift'. Width of EDZ up to about 1 m identified around the drift. **c** Flow-path sealing experiment using 'concretion-forming resin' injected in eight resin holes (no. 1–8) was carried out in the EDZ developed beneath the gallery floor. No. 5 and 6 resin injection holes were over-cored after 17 months. **d** A highly permeable zone occurs between the floor concrete and the bedrock observed by BTV (Borehole TV). All figures were created by H. Yoshida.

expected as long as the groundwater supplies the relevant ions. This method of stimulating calcite formation can be applied to achieve long-term sealing of natural and/or anthropogenically formed fluid flow paths around tunnels, caverns, and boreholes. Such synthetic sealing of flow paths will be maintained for as long as the surrounding geochemical conditions persist.

To examine the ability of the resin to seal bedrock flow paths, an in situ experiment has been conducted in an underground research laboratory (URL) located in Horonobe, northern Hokkaido, Japan (Fig. 2a)[40]. This URL has been constructed in the Wakkanai formation, a Tertiary diatomaceous siliceous rock without any carbonate cement or carbonate fracture-filling veins, but containing relatively bicarbonate-rich groundwater of sea-water origin[41,42]. The experiment is focused on sealing the EDZ developed around an underground experimental gallery excavated at a

depth of 350 m below the surface (Fig. 2b). The EDZ here was caused by the blasting method used to excavate the gallery and has been characterized by tomography[43]. The hydrogeological disturbance due to the EDZ has also been determined by borehole core observations, wireline logging, and several types of hydraulic tests[44]. Based on these investigations, the EDZ is well characterized and has been shown to extend into the rock for up to ca. one-meter depth from the gallery wall. In particular, the frequency of fractures in the EDZ is at least ten times higher than in the undamaged host rock[44] (Supplementary Note 2). Reflecting this difference in fracture frequency, the hydraulic conductivity of the EDZ is two to three orders of magnitude higher (in the order of $10^{-6}$ to $10^{-5}$ m/s) than the undisturbed host-rock (Fig. 2b, c)[45,46]. Consequently, the EDZ acts as a preferential groundwater flow path, together, with crushed rocks that fill the space between the floor

**Fig. 3 | Hydraulic conductivity change after 'con-cretion-forming resin' injection.** Measurements were carried out to trace the effect of the 'concretion-forming resin'. Earthquake shocks were identified and flow-path resealing was confirmed, with the hydraulic conductivity recovering rapidly within a few months after the shocks. After 14 months following the earthquakes, hydraulic conductivity has almost reached the level of undisturbed rock. EDZ excavated damaged zone, URL underground research laboratory.

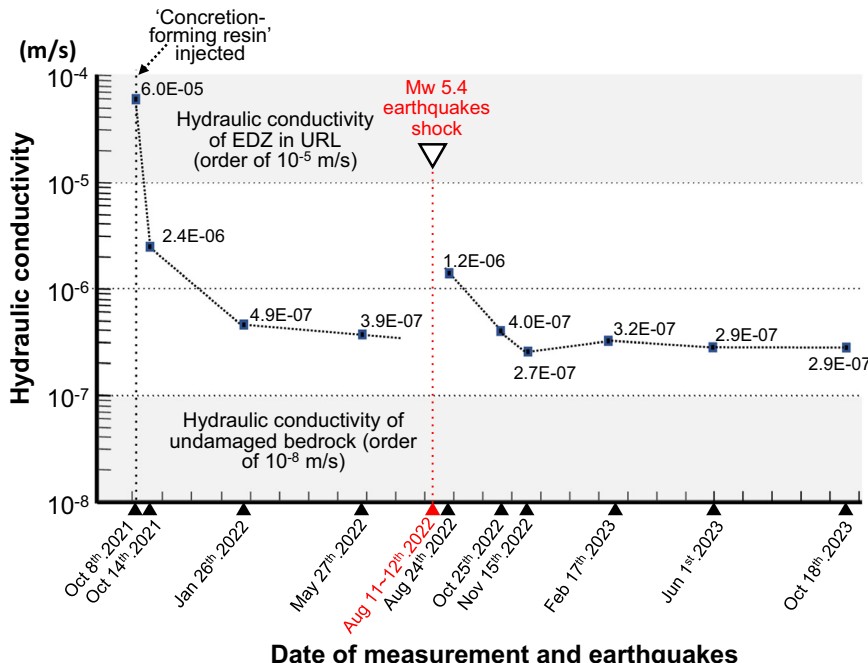

concrete of the gallery and bedrock (Fig. 2c, d). Such EDZ structural features have been confirmed by bore-hole television (BTV) observations carried out in all boreholes drilled (Fig. 2d). The thickness of the crushed rocks between the floor concrete and bedrock is about 10 cm and highly fractured EDZ bedrock is also readily identified. Characteristically the EDZ is also developed approximately parallel to the gallery wall and consists of a fracture network that provides a zone of relatively high permeability that acts as a groundwater flow path[44].

To measure the effectiveness of the 'concretion-forming resin' technology for sealing the flow paths in the EDZ hydraulic conductivity was monitored using hydraulic testing equipment comprising a single packer system in a central borehole (depth: 1 m; diameter: 50 mm) around which was located eight boreholes into which resin was injected (depth: 1.5 m; diameter: 50 mm) (Fig. 2c and Supplementary Note 2). This resin injection was undertaken using only a hand-operated machine thereby avoiding the use of a high-pressure mechanical system. The volume of the resin was almost the same as that of the drilling hole (ca. 4.4 kg of resin containing ca. 330 g of Ca in one resin hole). A constant pressure hydraulic test (Supplementary Note 3) was carried out every two to four months after the resin was injected, from August 2021 until October 2023 (almost 26 months). These tests enabled changes in the hydraulic conductivity of the EDZ to be measured.

During this period of monitoring, after 11 months of resin injection, unexpectedly six measurable inland earthquakes occurred within only two days (11th–12th of August, 2022). The maximum magnitude was *Mw* 5.4 on 11th August at 00:53 (at the URL site it was 5-upper on the Japanese earthquake intensity scale, equivalent to about VII on the Modified Mercalli Scale). The epicenters were at the URL site and the foci were at 2–7 km depth below the site (Supplementary Note 4). Due to the regional hydrogeological disturbance caused by the earthquakes, pore-water pressure at the 350-meter level in the URL increased by almost 75 kPa (Supplementary Note 4). In order to evaluate the durability of the sealing effect, post-earthquake in situ measurements were conducted continuously until October 2023.

After 17 months of monitoring (January 2022) from the beginning, two of the resin holes were over-cored (φ100 mm) and six new boreholes were drilled in order to identify the distance from the resin holes over which the flow paths in the EDZ were effectively sealed. All boreholes drilled were filled by mortar immediately after sampling the cores and monitoring of hydraulic conductivity was continued until October 2023.

Identification of calcite filling was carried out by core observations and BTV observations in the new boreholes (Supplementary Note 5). As noted above, the natural Wakkanai formation contains no calcite, so all calcite identified must have formed due to the use of the resin. Hydraulic conductivity was measured on an over-cored core after the 'concretion-forming resin' hardened, and the result was compared with the hydraulic conductivity of a pristine core of the Wakkanai formation (Supplementary Note 6). Detailed microscopic analysis by optical microscope and scanning electronic microscope (SEM) was also carried out to understand how calcite had formed and clogged the flow paths (Supplementary Note 7). Synthetically formed calcite and Ca concentration in and around the flow paths can be identified readily by X-ray diffraction (XRD) (Supplementary Note 8) and X-ray analytical microscopy (SXAM) (Supplementary Note 9), respectively because the rock does not contain calcium carbonate naturally. The Ca concentration in the concretion resin in the over-cored core was also measured to confirm the quantity of Ca consumed by seal formation during the 17 months of the test (Supplementary Note 10). The quantity of Ca remaining in the resin was used to estimate the additional period for which seal formation provided by the 'concretion-forming resin' will continue.

## Results

During monitoring, the 'concretion-forming resin' worked effectively to seal the EDZ flow paths, the hydraulic conductivity decreasing drastically to lower than 1/100 (from the order of $10^{-5}$ m/s to $10^{-7}$ m/s) of the initial value within a week to a few months after resin injection (Fig. 3 and Supplementary Note 3). Due to the earthquakes, the hydraulic conductivity of the EDZ increased by almost one order of magnitude. However, the hydraulic conductivity rapidly decreased again to the same level as before the earthquakes within a few months. Subsequently, after over-coring, the remaining resin in the six injection holes worked continuously to seal the flow paths. As a result by October 2023 (14 months after the earthquakes), the hydraulic conductivity had decreased further to close to the level of the undisturbed rock (in the order of $10^{-8}$ m/s). Such a sealing ability shows the continuous effectiveness of the resin even after damage caused by unavoidable geological perturbations.

Observations on over-cored cores, BTV observations, and XRD analysis revealed that all fractures seen in the EDZ were filled by calcite (Fig. 4a, b and Supplementary Notes 5 and 8). A cross-section through an over-cored core shows that 'concretion-forming resin' has filled the

**Fig. 4 | Calcite sealing identified in flow paths and host rock matrix promoted by 'concretion-forming resin'. a, b** All flow-path fractures are sealed by calcite identified in the rock core obtained by over-coring. **c** Calcite on a flow-path surface identified in the core drilled a distance of 20 cm from a resin injection hole no. 6. The surface of the fracture flow-path is covered only by calcite (there is no resin contributing to sealing) and the morphology of the calcite suggests the channel-like character of the flow-path. **d, e** SEM photos of calcite filling formed on the surface of the fracture flow path observed in (**c**). Euhedral calcite crystals (**e**) cover almost all the fracture surface thereby clogging the porosity of the fracture flow paths. **f** Core obtained by over-coring injection hole no. 6 shows calcite crystals have formed within the 'concretion-forming resin' thereby helping to maintain the durability of its sealing capability. EDZ excavated damaged zone.

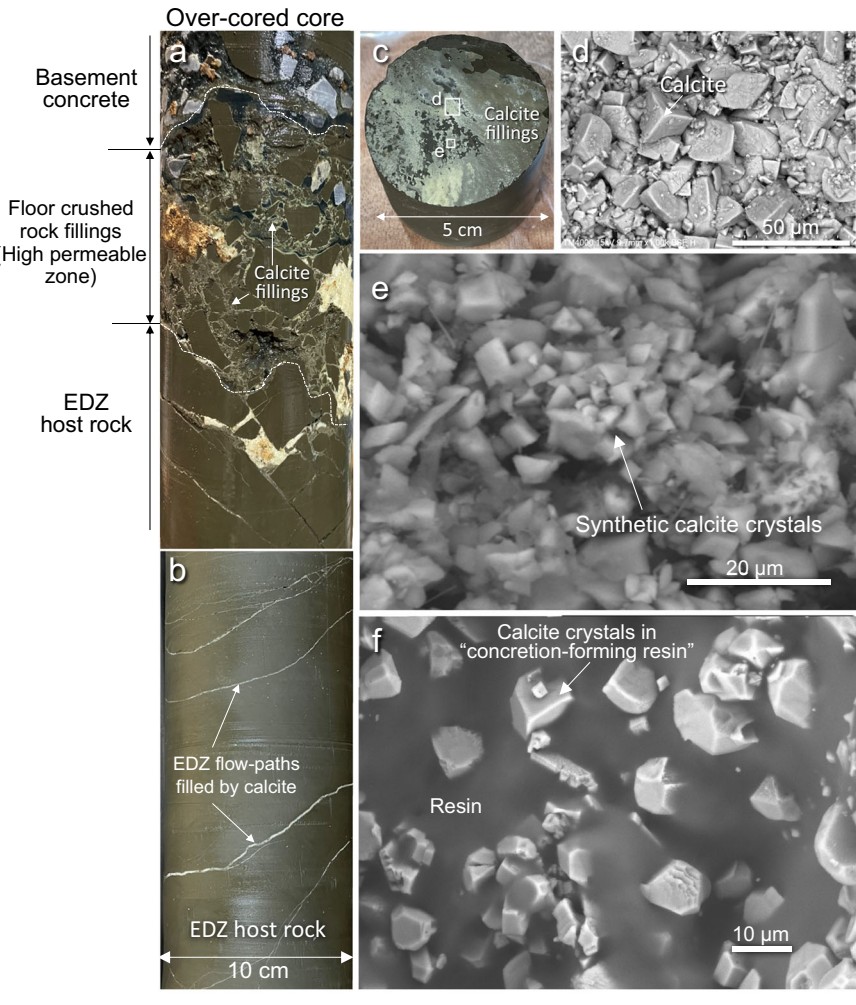

injection hole, but has not impregnated flow paths remote from the hole (Fig. 5a). Calcite fillings alone (i.e., without resin) were also identified in the fractures of cores obtained from holes drilled up to 20 cm from the resin injection hole (Fig. 4c, d). Tests of the over-cored core also showed quite a low hydraulic conductivity (order of $10^{-10}$ m/s) (Supplementary Note 6), almost the same as that of the pristine core from the Wakkanai formation. The decrease in hydraulic conductivity was due to the filling of porosity by calcite as observed by optical microscopy.

Observations by optical microscope also show that the open pore space originally in the flow paths has been clogged by euhedral calcite crystals. Micro-sized crystals were also observed in the rock matrix adjacent to the flow paths (Fig. 4e). SEM analysis confirmed that aggregates of euhedral calcite crystals with sizes of a few microns to tens of microns have grown on all seen fracture surfaces and filled flow paths (Fig. 4d, e and Supplementary Note 7). Such micron-sized euhedral calcite crystal formation is also seen in the 'concretion-forming resin' and no pore space was observed even after 17 months of in situ experiment (Fig. 4f).

Analyses by SXAM of Ca on the crosscut surfaces of over-cored cores indicate that in the rock surrounding the resin injection hole, all fractures and the rock matrix adjacent to these fractures have high Ca concentrations (Fig. 5a, b and Supplementary Note 9). These analyses show that Ca has migrated from the wall of the resin injection hole along fractures for distances of up to several cm, and diffused for a few millimeters into the rock matrix in the fracture walls. In the thin section, it is clear that the elevated Ca is not due to resin penetration, but rather to the precipitation of micron-sized calcite crystals that clog the matrix pores (Fig. 5c, d). A cross-section through a fracture shows that fine-grained calcite crystals grew first, then relatively coarser crystals developed afterward (Fig. 5e). Ca concentrations

measured in the resin (hole no. 6) after the experiment show that only ca. 20% of the $Ca^{2+}$ ion present initially was consumed during the 17 months of the experiment (Supplementary Note 10). Subsequent monitoring is left to evaluate the durable sealing effect provided by the remaining Ca ions present in the 'concretion-forming resin'.

## Discussion
### Sealing process by 'concretion-forming resin'
Based on the experiment, we found that the process of sealing and post-earthquake resealing of fracture flow paths and the adjacent rock matrix pores progressed by the formation of different progressively coarser calcite crystals (Fig. 5f). This occurred rapidly as a result of calcite super-saturation caused by the ions supplied by the hardened resin mixing with the solutes in the groundwater. In detail, the process of 'concretion-forming (calcite sealing)' is explained by the following three steps.

Step 1: hardening of the resin itself, takes a few tens of minutes to several hours. After hardening, concretion-forming Ca and $HCO_3$ seep out to produce calcite super-saturated conditions in the surrounding porewater, thereby promoting calcite precipitation (Step 2).

Step 2: calcite precipitation and sealing of flow paths and pores due to calcite super-saturation caused by high concentrations of $Ca^{2+}$ and $HCO_3^-$ originating in the resin (Step 1) adding to the $Ca^{2+}$ and $HCO_3^-$ in the groundwater. The process progresses within a few weeks to months, as confirmed both by the field experiment and a supporting laboratory experiment[32] (Supplementary Notes 10 and 11). This process is expected to proceed until the source ions in the resin are consumed. During this step, the concretion-forming ions diffuse through any fractures that form flow paths into the adjacent rock matrices (a process termed rock matrix diffusion[47,48]).

**Fig. 5 | SXAM mapping of a cross-cut through rock core obtained by over-coring. a** Cutting surface of the over-coring core (hole no. 6). **b** Ca mapping of rock core obtained by over-coring a resin injection hole indicates all observed fractures are sealed by calcite for up to several centimeters from the resin injection hole and that matrix pores in the walls of fractures are also calcite-filled. Squares indicate the observation points shown in (**c**–**e**). **c**, **d** Optical microscope view (cross-polar) showing calcite clogging fracture flow paths shown in the points of (**b**). Calcite clogs the flow paths in the rock matrix. Interconnecting calcite crystals effectively fill pore space even in hair cracks observed in the rock matrix (**d**). **e** SEM micrograph showing a cross-section through a calcite-filled flow path shown in (**b**) in which the calcite has a zonal growth texture. **f** Conceptual view of zonal clogging features observed in and around the flow paths far from the resin injection hole. Within flow paths far from resin injection holes, synthetic calcite forms on both fracture surfaces directly. Microcrystals grew first and larger crystals developed on the microcrystals to reduce the flow-path porosity. Calcite is also developed in the rock matrix near the fracture flow paths due to matrix diffusion of concretion-forming solutes derived from the resin, as shown by Ca maps produced by SXAM (**b**) and optical microscope (**c**). The color bar shows the X-ray intensity of measured Ca is shown. EDZ excavated damaged zone. All figures were created by H. Yoshida.

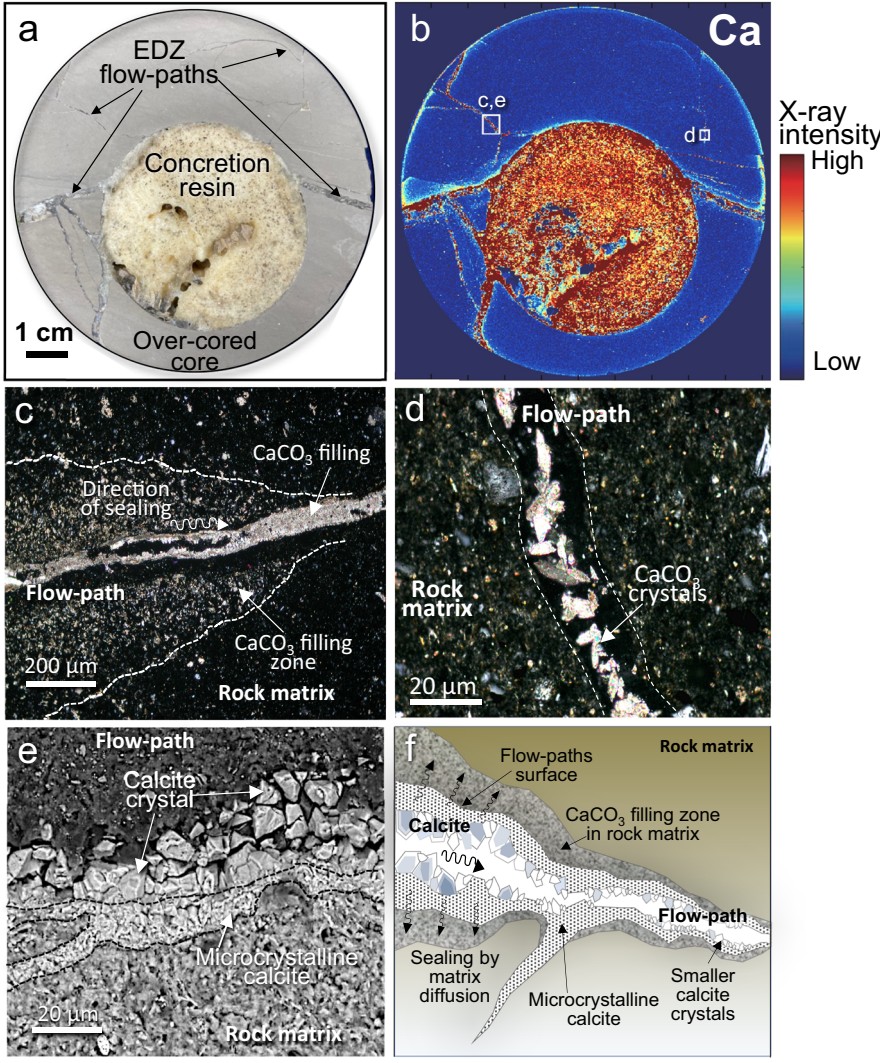

Consequently, not only fractures but also micro-pores in the rock matrices are clogged by calcite precipitation.

Step 3: After the synthetic calcite nucleation, calcite crystals formed initially will be further overgrown for as long as the ions ($Ca^{2+}$ and $HCO_3^-$) are supplied from the groundwater/porewater. Such a process is similar to the well-known formation of calcite overgrowths during diagenesis[49].

Although direct evidence is difficult to find, two reasons probably explain the initially increased hydraulic conductivity caused by the inland earthquakes: firstly the earthquake re-opened the flow paths which had been clogged by synthetically formed calcite, and secondly, the earthquake might have caused new transmissive fractures to open in the EDZ. In either case, irrespective of the cause of the increased hydraulic conductivity of the EDZ after the earthquakes, within a few months, the initial hydraulic conductivity was recovered due to additional precipitation of new calcite along the flow paths leading to their sealing. This sealing continued to decrease the hydraulic conductivity to close to the level of the undisturbed rock.

The calcite seen on the fracture surfaces in a core obtained 20 cm from the location of resin injection shows that calcite clogging occurred preferentially along the network of fracture flow paths in the EDZ. Optical microscopy and determinations of Ca distributions by SXAM also revealed that calcite formed not only in flow-path apertures, but also in the micro-pores of the rock matrix. The latter can be explained by the diffusion of Ca into the matrix driven by the steep concentration gradient caused by the resin releasing Ca to the water in the flow-path. These observations show that the 'concretion-forming resin' causes rapid flow-path sealing beyond the spatial limits of the 'concretion-forming resin' hardening. In other words, the seal by calcite clogging is more spatially extensive than the resin itself.

Furthermore, this sealing beyond the spatial limit of the 'concretion-forming resin' occurs after the resin has hardened owing to the release of Ca to groundwater by the resin. As this sealing is passive, caused by solute diffusion without pressurization, micro-pores are sealed without physical force, thereby preventing any physical damage to the rock and the generation of new flow paths (which then need to be sealed). The resin will maintain the capability to reseal any flow paths that might re-open as a result of processes like seismicity for as long as the resin is a source of concretion-forming ions. In the absence of such perturbing processes, once calcite has been formed, the flow-path seals will be maintained for as long as the surrounding geochemical conditions persist.

## Quantitative estimates of Ca distribution and sealing distance

The process stimulating calcite formation demonstrated in this study can be applied to achieve long-term sealing of natural and/or anthropogenically formed fluid flow paths around tunnels, caverns, and boreholes. Because calcite formation will continue until the concretion-forming ions in the resin are all consumed, the seal will remain durable until at least this time. Ca concentrations measured in the resin in the over-cored rock show that only ca. 20% of the $Ca^{2+}$ ion present initially was consumed during the 17 months of the experiment (Supplementary Note 10). The in situ Ca leaching rate (ca. 20% during 17 months) is slower than the rate measured in the laboratory (ca. 36% during 3 months; Supplementary Note 10). This difference can be explained by the laboratory measurement being carried out in free water, in

which diffusive transport of Ca from the resin surface will be faster than in the porous Wakkanai formation, in which diffusion follows a tortuous path through matrix pores or narrow fractures. Calcite formed in the laboratory experiment using distilled water, demonstrating that the sealing technology does not rely on the groundwater providing $Ca^{2+}$ and or $HCO_3^-$; both components are supplied by the resin. However, the existence of other chemical components in real groundwater, as well as the salinity of the groundwater, would influence the quantity and rate of calcite precipitation. This is a topic that requires further study.

In the investigated Wakkanai formation, natural groundwater flow has been shown to be extremely slow, with a component of palaeo-seawater still preserved in the porewater[50]. In the matrix of the Wakkanai formation, the transport of solutes will be diffusion-dominated[51]. The distance over which Ca may be transported by diffusion can be estimated using the following equation[52]:

$$L = \sqrt{2D \cdot t} \quad (1)$$

Where: $L$ is the diffusion length (m), $D$ is the effective diffusion coefficient ($m^2/s$) and $t$ is the time (s).

In the matrix of the Wakkanai formation $D = 10^{-10}\ m^2/s$ (based on through-diffusion experiments using non-sorbing tracers[51]). Equation (1) then gives a maximum distance of Ca diffusion through the matrix of about 10 cm over 17 months. On the other hand, diffusion in fracture flow paths would be faster than in the matrix because fractures have fewer constrictions than matrix pores and present a smaller surface area per unit volume of water on which solute-mineral interactions can occur. However, diffusion through the fractures is difficult to estimate, given that the fractures have narrow apertures and many asperities on their walls. An upper limit on the diffusion distance in open fractures can be estimated by using the free-water diffusion of approximately $1 \times 10^{-9}\ m^2/s$[53]. Using this value Eq. (1) gives a diffusion distance of about 40 cm after 17 months. This distance is larger than the observed maximum distance of 20 cm from the no. 6 resin injection hole where calcite has been observed (Supplementary Note 2) as shown in Fig. 4. However, it is possible that calcite has formed at greater distances than sampled by rock coring, while the estimated distance of 40 cm is an upper estimate and the actual diffusion distance is likely to be shorter, taking into account asperities in fracture walls and wall-rock solute interactions such as physicochemical and/or electrically induced adsorption by mineral surfaces. Considering these factors, the observations are consistent with expectations based on theory.

The estimated distance of sealing is based on data from just one resin injection hole. The same sealing effect would be expected for the other 7 resin injection holes. The distance between neighboring injection holes is ca. 20 cm. Therefore, the results of the diffusion calculation imply that the $CaCO_3$ clogging front from a resin injection hole may overlap with that from the neighboring resin injection hole. If a fracture flow path connects neighboring resin holes, then an overlapped sealing effect would occur.

The volume of open fracture porosity that would need to be sealed by calcite to produce the reduction in hydraulic conductivity observed during the in situ experiment (Fig. 3), is difficult to estimate. This is because the hydraulic conductivity of a fracture depends not only on its open, connected volume, but also on its tortuosity, variations in constrictivity, and morphology of the fracture surfaces[54,55], none of which can be measured easily. However, an upper estimate of the fracture volume that would need to be filled with calcite can be made using the cubic law, which is applicable to a fracture with parallel, smooth walls[54]:

$$K = \frac{\rho g b^3}{12\mu} \quad (2)$$

Where: $K$ is the hydraulic conductivity (m/s), $\rho$ is the density of the water ($kg/m^3$), $g$ is the acceleration due to gravity ($m/s^2$), $b$ is the fracture aperture (m), and $\mu$ is the dynamic viscosity of water ($Ns/m^2$).

During the in situ experiment, the hydraulic conductivity of the rock decreased from $6 \times 10^{-5}\ m/s$ to $3.2 \times 10^{-7}\ m/s$ (Fig. 3). If the flow was through a single fracture per $m^2$ cross-section of rock, and assuming that the water has a density of $1000\ kg/m^3$ and viscosity of $0.001\ Ns/m^2$, then according to Eq. (2) the aperture would decrease by 0.35 mm. The radial distance over which such a reduction in aperture might occur can be estimated using the following equation:

$$D = \sqrt{V/\pi \cdot W} \quad (3)$$

Where: $V$ is the volume reduction ($cm^3$), $W$ is the aperture reduction (cm), and $D$ is the radial distance from the resin injection hole over which the volume reduction occurs.

During the experiment, about 20% of Ca in the resin, about 66 g, was leached from one resin hole (Supplementary Note 10). If all this leached Ca precipitated as $CaCO_3$, about 165 g of $CaCO_3$, or ca. 60–70 $cm^3$ by volume, would form. Taking the lower value as $V$ in Eq. (3) and the aperture reduction calculated above as $W$ in Eq. (3) gives a distance $D$ of about 24 cm after 17 months. This distance is comparable to the maximum observed distance from the injection hole at which calcite is observed.

In reality, the measured water flow occurred through more than one fracture, as shown by the calcite-filled fractures seen in the core (Fig. 4). Hence, the cubic law will over-estimate the actual permeability and far less calcite precipitation will be needed to decrease the permeability of a real fracture with rough walls that come into contact over a significant proportion of the fracture. It is plausible that such an overestimate could be by an order of magnitude or more[56].

BTV observation and examination of the over-cored core show that in the Wakkanai formation, the fractures generally have apertures <1 mm (Figs. 2 and 4) and also that not all fractures are open. The features indicate that the accessible open space and clogging of calcite would be limited (Fig. 4c) if we consider both effects on the channeling of a fracture surface and the actual flow path having a smaller aperture.

Considering the above factors, the calculated volume of calcite formed by precipitation of Ca released from the resin is consistent with the observed decrease in hydraulic conductivity.

## Advantages compared to conventional methods

The relatively small proportion of Ca leached from the resin (ca. 20%) during the 17 months of the experiment also implies that the progressive sealing function provided by the 'concretion-forming resin' will continue for at least 4–5 times longer than the duration of monitoring. However, the early-formed calcite will reduce the diffusivity of the rock by clogging pores, and therefore reduce the rate at which $Ca^{2+}$ and $HCO_3^-$ can diffuse from the resin. It is therefore reasonable to expect that the ability of the resin to promote additional clogging in the event of perturbations will be many times longer.

The durability of the seal can be controlled by adjusting the volume of injected resin, which will be achieved by varying the diameter, number, layout, and lengths of injection holes. Additionally, if the groundwater contains $Ca^{2+}$ and $HCO_3^-$ ions, there will be increased calcite supersaturation leading to a further positive effect on the rate and volume of pore-clogging by calcite formation. Once the calcite has formed in the pores it will persist or overgrow in remaining pores for as long as the geochemical conditions are favorable (calcite saturation or supersaturation).

Deep underground bedrocks are commonly saturated by groundwater in which calcium and bi-carbonate ions are dissolved and pH is neutral to alkaline[57,58]. Hence, our newly developed sealing technique can be applied to many types of rocks during various underground activities that require long-term containment and/or isolation of materials.

Although the ability of the resin to sorb solutes such as radionuclides is not well analyzed, the physical heterogeneity of the resin surface is expected to favor sorption[59]. Also, calcite formation will be effective in co-precipitating certain solutes, notably Sr, during pore clogging. This

process may work to retard the migration of the co-precipitated solutes[60,61]. Even if the resin has a low sorption capacity, it will effectively work as a physical barrier to solute migration due to the resin's low solubility in groundwater.

The methodology can also be used to complement cementitious seals, taking advantage of the elevated $Ca^{2+}$ concentrations in leachates from cementitious materials. Again, these elevated $Ca^{2+}$ concentrations work in combination with the ions produced by the resin to cause enhanced supersaturation of calcite and hence more rapid calcite precipitation and consequent flow-paths sealing. The sealing processes at the heart of this new technology are quite different from those in conventional grouting methods using cementitious materials that are currently used. Furthermore, the seals produced by the new technology are expected to be more durable.

## Conclusion

To use of 'concretion-forming resin' (what we call 'concretion-seeds') led to the rapid sealing of the fracture network developed in the EDZ around a tunnel at 350 m depth by the formation of synthetically formed calcite. Sealing was clearly identified by in situ hydraulic tests, core observations after monitoring, and detailed mineralogical and geochemical analysis. Since the studied Wakkanai formation does not contain calcite, all the observed calcite in the rock core sampled after the in situ experiment to test the sealing technology must have formed as a result of Ca being released by the resin and reacting with groundwater constituents. Both mass balance and diffusion calculations provide approximate estimates of the rock volume in which fracture sealing by calcite occurred. These estimates are broadly consistent with the observations.

The seals formed by the calcite self-healed rapidly after physical damage caused by earthquakes. The earthquakes caused an initial increase in rock permeability, but this then decreased to a level similar to the natural rock matrix owing to continued calcite precipitation around the resin. The EDZ flow paths sealed in the present study were fractures formed synthetically during underground drift construction. However, these fractures behave as flow paths similar to natural fracture networks that occur in many kinds of rock. The demonstrated effective sealing of the EDZ fractures by the 'concretion-forming resin' therefore shows that this resin may be used also to seal natural fracture networks.

The resilience of the seals after earthquake shocks in the study reported here shows that seals produced by the new technique will be resilient after other kinds of unavoidable perturbations that may occur underground. Examples of such perturbations are tunnel convergence, cyclical pressurization/depressurization during gas storage cavern operation, or thermal shock caused by injecting fluids through boreholes in underground usage.

Many groundwaters are naturally in a state of chemical saturation with respect to calcite, owing to the widespread occurrence of this mineral in many rocks. It is to be expected that seals developed in such rocks will be effectively permanent. The preservation of natural spheroidal calcite concretions for many millions of years is an analog for this permanent sealing.

## Methods

### Hydraulic conductivity test of EDZ after 'concretion-forming resin' injection

A steady-state hydraulic conductivity test with a packer was located in a borehole that had been drilled through the concrete floor of the tunnel into the EDZ (Supplementary Notes 2 and 3). Water was injected into the test section while the effective water injection pressure was increased in stages, with a steady flow rate being achieved in each stage. Using the measured steady-state flow rate, the coefficient of hydraulic conductivity was calculated. For this test of the hydraulic conductivity of EDZ and rock mass, a low effective water injection pressure was used to ensure that no deformation or collapse of the flow paths or pore structures occurred in the EDZ. The results of the test are summarized in Supplementary Note 3.

### Calcite identification after the experiment

BTV observations were carried out in the over-cored boreholes to identify flow paths sealed by calcite due to injection of the concretion-forming resin (Supplementary Note 5). Detailed morphological occurrence of synthetically formed calcite in flow paths was characterized by SEM (S-3400N, Rigaku Co. with 15 kV acceleration voltage) of fillings taken from the overcoring core (Supplementary Note 7). The mineralogical compositions of the fracture fillings were also determined with an X-ray diffractometer (XRD; Multiflex, Rigaku Co.) using crushed and powdered samples and Cu Kα radiation (the Cu being subjected to an electron beam of 40 kV/20 mA) (Supplementary Note 8).

### SXAM for Ca concentration measurement

Cross-sections of over-cored core were used to determine Ca concentration profiles across the concretion-forming resin and surrounding rock matrices. Ca concentrations were determined using an X-ray fluorescence analyzer (SXAM: XGT-5000V Horiba Japan) at the Nagoya University Museum, Aichi, Japan. SXAM intensity maps of over-cored cores were reduced to one-dimensional element profiles in a direction perpendicular to the concentric rings formed by variations in the concentrations of Ca and other elements. The measurements were made by focusing a high-intensity continuous X-ray beam (Rh anode 50 kV 1 mA), 10 μm in diameter, using a guide tube oriented perpendicular to the sample surface. The tube was located on a PC-controllable X–Y stage. X-ray fluorescence from the sample surface was analyzed with the hp-Si detector of an energy-dispersion spectrometer[62]. Other elements measured semi-quantitatively were Fe, Mn, and K across the whole surface, as shown in Supplementary Note 9.

### Ca concentration measurement of resin before and after the experiment

The Ca concentration of the concretion-forming resin in the over-cored core was also measured to confirm the quantity of Ca consumed during the experiment. The purpose was to estimate the period for which the sealing process would continue. Ca concentrations were measured by using atomic absorption spectrometry (Hitachi A-2000) with the following procedures. The concretion-forming resins (the original one and the resin remaining after the experiment), were decomposed by heating at 200 °C (15 min.), increased to 300 °C (15 min.), and subsequently to 400 °C (30 min). The Ca in the residue was dissolved in 2%-$HNO_3$ and diluted to 10–20 ppm. Standard solutions of 5, 10, 20, and 30 ppm were used for calibration. The results are shown in Supplementary Note 10.

## Data availability

All data needed to evaluate the conclusions in the paper are present in the paper and/or the Supplementary materials. Additional data related to this paper may be requested from the authors.

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

## Acknowledgements

We thank the JAEA Horonobe URL site staff for their collaboration work in underground, as well as for their helpful advice during the experiment. We also thank Taisei Corporation for safety control during the in situ experiment. In situ hydraulic measurement is supported by Asano Taiseikiso Engineering Co., Ltd. Laboratory permeablity measurement is supported by M. Ishibashi of Kajima Corporation. We are grateful to S. Sirono, N. Takagi of Nagoya University, S. Nishimoto of Aich University, and S. Takeuchi for their technical discussion and support in preparing rock thin-sections, SXAM mapping and SEM-EDS, and XRD analyses, respectively. The research was supported by the following funds; Chubu Electric Power Company and METI researcher fund (JPJ007597), as well as JSPS KAKENHI grant 18H052227 and 23H00280.

## Author contributions

All authors jointly planned and designed the research. H.Y. and R.M. wrote the original draft, and all authors contributed substantially to revisions. K.Y., Y.A., I.M., A.U., and N.K. analyzed the data, and all authors interpreted the results. K.K. and A.S. prepared the concretion resin. H.M., A.M., and M.J. executed an in situ experiment and the safety control.

## Competing interests

Tokai National Higher Education and Research System (H.Y.) applied for patents related to the technology described in this work. Title: structure material, structure, method for manufacturing structure, seal composition, and ion supply material. Application no.: CN 201980054826.7, JP 2020538460, US 17/270,297. Tokai National Higher Education and Research System has granted a company a non-exclusive license to exploit the application (JP 2020538460). All other authors declare no competing interests.
