## [Peer Review File · Communications Engineering]

Reviewers' comments:

Reviewer #1 (Remarks to the Author):

The authors presented a work on "Post-earthquake rapid resealing of bedrock flow-paths by 'concretion-forming resin'." The work highlights the challenge of sealing flow paths after an earthquake and criticizes the impractical solutions proposed in the study. I agree with the idea of addressing post-earthquake issues, as there are several concerns related to identifying flow paths and implementing suitable technology for accessing those areas.

In the context of reservoirs, injection and production wells serve as conduits to reach reservoirs that are typically situated away from faults and fractures. However, in the aftermath of an earthquake, separate drilling may be necessary to establish new wells. If the proposed solution involves utilizing injection wells, it could potentially block all the pores and pore channels, which is why I do not recommend considering this approach for CO₂ storage.

Nevertheless, this concept could find utility in the field of civil engineering, where it can be applied to address other challenges unrelated to carbon capture and storage (CCS).
At this moment, I don't consider it for publication.

Reviewer #3 (Remarks to the Author):

The paper described a new resin-based grouting technology that mimics the natural formation of calcite concretions for applications in rock fracture grouting. The research is very interesting and the idea is novel. The authors show that over a period of several years, calcite crystals progressively form as Ca and HCO₃ diffuse from the newly developed resin causing Ca saturation and precipitation of calcium carbonate on rock surfaces.

The paper is very interesting and I would like to see the technology published. However, there is a complete lack of information regarding the new resin technology itself. I appreciate that the authors cannot provide detailed commercial information. However, some further information on the resin chemical composition and hydraulic properties is required to understand the process.

During the field experiment, conducted over several years, not all of the available Ca in the resin is used. Is this because the formation of CaCO₃ inhibits further interaction between the resin and the groundwater (i.e. the calcite blocks access to groundwater at the grout front)? Or is it because the groundwater is saturated with respect to Ca? Since there is little Ca in the natural groundwater, I would expect the Ca concentration to drop quite rapidly due to diffusion. Thus, it tends to imply that Ca forms on the resin surface i.e. the interface between the resin and the groundwater. Do you have any evidence for this?

Does the CaCO₃ form directly on the rock fracture surfaces or is there resin in-between? Again do you have SEM images and EDX profiles that can demonstrate that calcite crystals nucleate and grow on the

fracture surfaces?

How quickly do you expect the calcite to form based purely on the permeability/diffusivity of Ca within the resin? How does this compare to the reduction in Ca in your cored sample?

How permeable is the resin? I presume that the initial drop in fracture permeability is in fact due to resin formation blocking the fracture aperture and not due to calcite formation? Where is the calcite forming with respect to the resin? Can you provide some evidence on the SEM images?

What remains of the resin once the Ca and HCO₃ have entirely diffused into the groundwater? What hydraulic properties does this remaining resin have? Or does the resin completely dissolve?

When the resin is initially injected, does it entirely fill the void space? If so, is the access to the surrounding groundwater, in order for CaCO₃ to form, limited only to the grout front? If so, this would imply that the CaCO₃ crystals only form beyond the grout front and not within the initial volume that is "resin-grouted". A cartoon that shows the geometry of CaCO₃ crystals, resin within a fracture, supported by evidence from the SEM data, would be beneficial in explaining the process.

Can you provide some information on the resin composition? How will the resin affect the chemical speciation/solubility of key radionuclides for example? Will radionuclides interact with other chemicals within the resin? These are questions that will inevitably be asked if this product is to be applied within a repository environment; long-timescale deterministic models of the chemical degradation and interactions of the product will be required for an environmental safety case.

Finally, can you comment on the evidence for calcite healing being the cause of the permeability reduction after the earthquakes? Do you see any evidence for cracking and resealing of the fractures fill within your core? For example, do you see cracking of the remaining resin itself, followed by calcite infilling the crack? Have you looked for this within the SEM data?

In summary, the research looks novel and interesting. However, some more information is required to understand (and hence properly review) the process and to determine its suitability for grouting in repository scenarios.

13th/Nov/2023

Dear Editor of *Communications Engineering*,

Answer to reviewers

Reviewers' comments:

Reviewer #1 (Remarks to the Author):

The authors presented a work on "Post-earthquake rapid resealing of bedrock flow-paths by 'concretion-forming resin'." The work highlights the challenge of sealing flow paths after an earthquake and criticizes the impractical solutions proposed in the study. I agree with the idea of addressing post-earthquake issues, as there are several concerns related to identifying flow paths and implementing suitable technology for accessing those areas.

In the context of reservoirs, injection and production wells serve as conduits to reach reservoirs that are typically situated away from faults and fractures. However, in the aftermath of an earthquake, separate drilling may be necessary to establish new wells. If the proposed solution involves utilizing injection wells, it could potentially block all the pores and pore channels, which is why I do not recommend considering this approach for CO₂ storage.

Nevertheless, this concept could find utility in the field of civil engineering, where it can be applied to address other challenges unrelated to carbon capture and storage (CCS). At this moment, I don't consider it for publication.

► The paper mentions CCS only as one of several possible example applications of the technology. The review comment seems to apply just to CO₂ injection wells. Actually, numerous studies have shown that the risk in many CCS projects is leakage from legacy wells. It is unclear how using a resin-based approach to seal these would be detrimental to a CO₂ storage project. Nevertheless, we wish to avoid distracting readers from the key purpose of the paper, which is to present a new technology for emplacing resilient seals in flow-paths through rocks, whether in boreholes or in engineered damages zones around excavations. Therefore, we have removed the mention of CCS to avoid possible confusion and focus instead on HLW waste disposal and other types of civil engineering usage.

Reviewer #3 (Remarks to the Author):

The paper described a new resin-based grouting technology that mimics the natural formation of calcite concretions for applications in rock fracture grouting. The research is very interesting and the idea is novel. The authors show that over a period of several years, calcite crystals progressively form as Ca and HCO₃ diffuse from the newly developed resin causing Ca saturation and precipitation of calcium carbonate on rock surfaces.

The paper is very interesting and I would like to see the technology published.

However, there is a complete lack of information regarding the new resin technology itself. I appreciate that the authors cannot provide detailed commercial information. However, some further information on the resin chemical composition and hydraulic properties is required to understand the process.

- ▶ As the reviewer has requested, information about the resin chemistry and hydraulic properties have been added to the text.

During the field experiment, conducted over several years, not all of the available Ca in the resin is used. Is this because the formation of CaCO_3 inhibits further interaction between the resin and the groundwater (i.e. the calcite blocks access to groundwater at the grout front)? Or is it because the groundwater is saturated with respect to Ca? Since there is little Ca in the natural groundwater, I would expect the Ca concentration to drop quite rapidly due to diffusion. Thus, it tends to imply that Ca forms on the resin surface i.e. the interface between the resin and the groundwater. Do you have any evidence for this?

- ▶ After running the experiment for 17 months, we have confirmed by over-coring that CaCO_3 precipitated as far as 20 cm from the resin injection hole. This has been described in the text. We think that this observation is concrete evidence of CaCO_3 sealing without CaCO_3 formation being inhibited by interaction between the resin and the groundwater (i.e. no calcite blocks the grout front).
- ▶ The release of Ca ions from the resin will be controlled by the solubility of the Ca seeds in the resin, such that calcite saturation is maintained in the flow-paths and pores, as situation that will continue until the Ca source has been completely consumed. Seventeen months from the start of the experiment, over-coring was conducted to check how much of the resin source had been consumed. The result shows that only about 20% of the source was consumed to form the calcite that clogs the surrounding flow-paths. Thus, the formation of calcite in the flow-paths was not inhibited by calcite formation on the resin surface, but by the seepage rate of Ca from the resin.
- ▶ Ca in groundwater can be also contribute to forming calcite in the flow-paths. If the groundwater contains Ca ions, calcite saturation will occur even more readily leading to calcite precipitation. Ca ions will be also be supplied continuously by the resin to form calcite until the Ca in the resin is consumed. We have confirmed that the calcite crystals are formed on and in the resin by SEM. An additional figure (Figure 5) has been added to the paper.

Does the CaCO_3 form directly on the rock fracture surfaces or is there resin in-between? Again do you have SEM images and EDX profiles that can demonstrate that calcite crystals nucleate and grow on the fracture surfaces?

- ▶ Optical microscopy, SEM observations and SXAM profiles show that CaCO_3 grows directly on the rock fracture surfaces. We have added a new figure to the paper to show this.

How quickly do you expect the calcite to form based purely on the permeability/diffusivity of Ca within the resin? How does this compare to the reduction in Ca in your cored sample?

- ▶ Calcite formation and cementation over time have been confirmed by laboratory experiments using concretion-forming resin, distilled water and glass beads. The results show that initially calcite formed quite rapidly (within a week) and that pores between the beads were almost filled after a few months. The process and rate are quite consistent with the reduction of Ca in the resin used in the in-situ experiment, as shown in the SEM figures explained above. We think that the same sealing process as observed in the laboratory experiment has occurred in the in-situ test.

How permeable is the resin? I presume that the initial drop in fracture permeability is in fact due to resin formation blocking the fracture aperture and not due to calcite formation? Where is the calcite forming with respect to the resin? Can you provide some evidence on the SEM images?

- ▶ It is important to point out that a key feature of our new technology is that immediate sealing is indeed caused by the resin, but this is enhanced by subsequent calcite formation (which occurs rapidly). Crucially, the calcite formation makes the seal more robust, because calcite is typically a stable phase in most deeper groundwater systems.
- ▶ The permeability of the resin is quite low ($<10^{-10-11}$ m/s has been measured in the lab.) Also the measured permeability of the over-cored core is in the order of 10^{-10} m/s. An explanation and data have been added to the text and supplementary.
- ▶ We think that the initial drop of permeability is indeed due in part to the effect of resin hardening, but the permeability drop is enhanced by calcite formation, which occurs within a week or so, leading to clogging of the flow-paths, as explained above. SEM images of showing calcite formed in the resin have also been added to the Supplementary.

What remains of the resin once the Ca and HCO_3^- have entirely diffused into the groundwater? What hydraulic properties does this remaining resin have? Or does the resin completely dissolve?

- ▶ As explained in our responses to the earlier comments, the resin remains even after the in-situ experiment, because of its very low solubility and the component is almost the same as 'amber'. The permeability was not also changed, during the in-situ experiment. In the longer-term, eventually the resin could degrade, but the duration of the time will depend on the precise conditions in which the resin is employed. The point of our work is that the seal does not depend on the longevity of the resin, but on the longevity of the calcite that forms as a result of the resin degradation. That is, in the short term the resin will contribute significantly to sealing. In the longer term its contribution will be less, but the seal will still be durable because of calcite

formation.

When the resin is initially injected, does it entirely fill the void space? If so, is the access to the surrounding groundwater, in order for CaCO_3 to form, limited only to the grout front? If so, this would imply that the CaCO_3 crystals only form beyond the grout front and not within the initial volume that is "resin-grouted". A cartoon that shows the geometry of CaCO_3 crystals, resin within a fracture, supported by evidence from the SEM data, would be beneficial in explaining the process.

- ▶ The liquid resin used in the experiments described in the paper is able to fill the void space, but not entirely because it has higher viscosity than water. Therefore, there are still some remaining voids, and solutes can diffuse through the void-filling water to access the resin. Due to this, CaCO_3 will form not only at the resin front but also in flow-paths far from the resin front, as well as inside the resin. We have added an additional explanation in the text and SEM images in the Supplementary.

Can you provide some information on the resin composition? How will the resin affect the chemical speciation/solubility of key radionuclides for example? Will radionuclides interact with other chemicals within the resin? These are questions that will inevitably be asked if this product is to be applied within a repository environment; long-timescale deterministic models of the chemical degradation and interactions of the product will be required for an environmental safety case.

- ▶ We have added details of the basic chemical components of the resin to the text. The resin is composing mainly of C,H,N,O that forms a polymer. Direct information concerning the sorption capacity of the resin for radionuclides is unavailable because experiments have not yet been undertaken. However, some sorption capacity may be expected due to the physical properties of the resin surfaces (we have also added a reference). We can be confident that given its low permeability and low solubility, the resin would work as a stable physical barrier rather than having a chemical barrier function. Before the resin could be used where radionuclides may contact it, additional work would be needed to establish that there would be no detrimental chemical interactions that might enhance radionuclide transport (e.g. complexing of radionuclides with resin degradation products).

Finally, can you comment on the evidence for calcite healing being the cause of the permeability reduction after the earthquakes? Do you see any evidence for cracking and resealing of the fractures fill within your core? For example, do you see cracking of the remaining resin itself, followed by calcite infilling the crack? Have you looked for this within the SEM data?

- ▶ We also think that this point is very important and we have made microscopic observations to look for evidence of fracturing or re-opening of flow-paths due to earthquakes. Although the identification of specific flow-paths that have been affected by an earthquake is quite difficult, we think that all flow paths being shown to be sealed by calcite after later

over-coring is quite compelling evidence. In the text, we have added a more detailed explanation about of possible reasons why the permeability was increased after the earthquakes.

- ▶ We also point out that the bedrock at the study site does not contain any calcite. Therefore, we can be confident that the calcite we see after the in-situ experiment has been formed by interactions between the groundwater and the resin.

In summary, the research looks novel and interesting. However, some more information is required to understand (and hence properly review) the process and to determine its suitability for grouting in repository scenarios.

Reviewers' comments:

Reviewer #3 (Remarks to the Author):

General:

It would have been much easier to review this, if the rebuttal letter had directed me to the corrected line and page numbers, so I did not have to hunt for the response to each of my questions in the text. In future, the authors should provide this.

Because, I have had to hunt for the appropriate text, my comments below are not in chronological order.

The paper is much improved, but there is still a lack of rigour in the reporting and a lack of quantification of Ca diffusion rates in the field trial, in comparison to expectations from laboratory data based on the resin. The question about whether calcium is being lost from the resin in the field at a slower rate than would be expected, based on the chemical properties of the resin, remains essentially unanswered.

Specific Comments:

Line 32 – p3. This is not an appropriate reference. 'Environmentally friendly' needs to be replaced with a quantifiable measure and some reliable data or references to support it. What do you mean? The web page is in Japanese and does not constitute a proper reference. If there is no reference, you should include supporting evidence in the Supplementary Information (SI).

P3, Lines 36-37. Are you able to simply quantify/model the diffusion rate and the fracture clogging? This would allow you to better understand the depletion in Ca in the resin over the 17 month-period and to predict the time taken for all of the Ca to diffuse. If you can show that diffusion of Ca from the resin is decreasing significantly over time, due to lack of direct contact with groundwater as the fractures clog, this would be beneficial. If this is the case, then the capability to repair subsequent cracks, could last orders of magnitude longer than you are anticipating (you currently say 4-5 times longer than the 17-month test).

Rebuttal letter – "CaCO₃ precipitated as far as 20 cm from the resin injection hole. This has been described in the text. We think that this observation is concrete evidence of CaCO₃ sealing without CaCO₃ formation being inhibited by interaction between the resin and the groundwater (i.e. no calcite blocks the grout front)."

Why do you think that? See above - have you modelled the process? It would be possible to calculate the mass of Ca expected to diffuse over 17 months, and thus, to model Ca concentration in the groundwater.

Surely the resin itself penetrates further than the injection hole? I would have thought that the resin could have penetrated 20cm. The question asked was whether you have CaCO₃ precipitation at the resin front (i.e. the furthest extent of the resin). Your response does not make this clear – do you know where the front is? i.e. what is the radial extent of the resin from the injection hole – do you know? You must know how much resin you injected and what the volume of the original injection holes was. You could use this to calculate the volume of resin that has penetrated into the rock fractures at least.

Which borehole did you over-core (on Figure 2) and where did you drill the hole that is 20cm away? It is not clear. I presume it is one of the core on SI Fig 3, but it doesn't say.

In the new Figure 5, where are images 5c, d, and e on Figure 5a? It is not clear where each sample is located. Figure 5f is really helpful, thank you. Where is the process described in 5f is located with respect to the injection hole? The resin itself is not marked on 5f. Is this a cartoon of processes beyond the volume into which the resin has penetrated?

Where the resin is present in the fractures, does it coat the fracture surfaces?

In Figure 5d, is it resin that surrounds the CaCO₃ crystals, or is it void space in the fracture? It isn't labelled.

Where are Figure 4c, d, e and f taken from? It is not clear whether these images are actually within the injection hole (now filled with resin) or whether they are located in the over-cored rock that surrounds it. For example, you show evidence in Figure 4 for CaCO₃ crystals within the resin, but it is not clear where the image was taken from. Is the resin within the injection hole or is it taken from a fracture in the overcored rock volume?

Response letter: "Calcite formation and cementation over time have been confirmed by laboratory experiments using concretion-forming resin, distilled water and glass beads...We think that the same sealing process as observed in the laboratory experiment has occurred in the in-situ test".

Where is the evidence for this in the paper or in the references? The letter doesn't give any clues. I presume it is Figure 12 in the SI.

What is the mass of calcite precipitated, with respect to the mass of resin in Fig 12 over the 3 month period. How does this compare to the Ca mass loss rate in the resin in the field trial? Is the field trial slower, or as expected?

Dear Editor of *Communications Engineering*,

We have no comments for the Editor.

Answer to reviewers

Reviewers' comments:

Reviewer #3 (Remarks to the Author):

General:

The paper is much improved, but there is still a lack of rigour in the reporting and a lack of quantification of Ca diffusion rates in the field trial, in comparison to expectations from laboratory data based on the resin. The question about whether calcium is being lost from the resin in the field at a slower rate than would be expected, based on the chemical properties of the resin, remains essentially unanswered.

► It is technically very difficult to determine the rates of Ca diffusion under in-situ conditions. However, it is not necessary to do so to demonstrate the efficacy of the methodology. While, we cannot provide a rate under in-situ conditions we are confident that the rate would not be faster than under lab conditions because clearly in-situ precipitation of the released Ca has decreased porosity and therefore transport rates of Ca must be reduced. This will be beneficial to provide a longer sealing effect than lab experiment. Thus, our conclusions regarding the timescales over which the resin may act to self-seal are pessimistic (conservative). The following specific comments address this comment more thoroughly.

Specific Comments:

Line 32 – p3. This is not an appropriate reference. 'Environmentally friendly' needs to be replaced with a quantifiable measure and some reliable data or references to support it. What do you mean? The web page is in Japanese and does not constitute a proper reference. If there is no reference, you should include supporting evidence in the Supplementary Information (SI).

► Although the resin itself does not contain any toxic and heavy metal ions or toxic volatile components, clear evidence that it is 'Environmentally friendly' has not been published yet. We therefore deleted this part from the text (Line 32-33 – p3 was deleted). We point out that the web page, though in Japanese, contains the Chemical Data sheet for the resin and as such is a valid reference.

P3, Lines 36-37. Are you able to simply quantify/model the diffusion rate and the fracture clogging? This would allow you to better understand the depletion in Ca in the resin over the 17 month-period and to predict the time taken for all of the Ca to diffuse. If you can show that diffusion of Ca from the resin is decreasing significantly over time, due to lack of direct contact with groundwater as the fractures clog, this would be beneficial. If this is the case, then the capability to repair subsequent cracks, could last orders of magnitude longer than you are anticipating (you currently say 4-5 times longer than the 17-month test).

► As mentioned in the reply to the general comment shown above, although the modelling of the fracture clogging has not been done owing to the technical difficulty of obtaining reliable data to support it, we also think that flow-path clogging will decrease the Ca release rate. This will result in the resin being effective for sealing flow paths over an even longer period

than the time we estimate. It was because we could not estimate reliable realistic in-situ leaching rates that we adopted a pessimistic (conservative) approach. We plan further work in an attempt to obtain more realistic data, but in the meantime the lack of this data does not invalidate the evidence for the resin being effective. Further explanation has been added in Line 37-38 – p3. Also related explanation is in the text (see in Line 24-28 – p8).

Rebuttal letter – “CaCO₃ precipitated as far as 20 cm from the resin injection hole. This has been described in the text. We think that this observation is concrete evidence of CaCO₃ sealing without CaCO₃ formation being inhibited by interaction between the resin and the groundwater (i.e. no calcite blocks the grout front).”

Why do you think that? See above - have you modelled the process? It would be possible to calculate the mass of Ca expected to diffuse over 17 months, and thus, to model Ca concentration in the groundwater.

► It is not necessary to model the process to draw this conclusion since it is supported by direct observations. Calcite has clearly been precipitated 20 cm from the resin injection hole in locations without any resin present. The natural rock itself contains no calcite so we can be quite sure that the calcite is formed by precipitation of constituents originating in the resin. Since there is no resin present, these constituents must have been transported beyond the resin front which cannot therefore have been blocked by calcite formation. We agree, however, that modelling of the process will be beneficial when planning future applications of the resin for sealing purposes. For this reason, our on-going research aims to obtain the data needed to support development of such a model. However, in the meantime the lack of such a model does not invalidate the conclusions of the paper.

Surely the resin itself penetrates further than the injection hole? I would have thought that the resin could have penetrated 20cm. The question asked was whether you have CaCO₃ precipitation at the resin front (i.e. the furthest extent of the resin). Your response does not make this clear – do you know where the front is? i.e. what is the radial extent of the resin from the injection hole – do you know? You must know how much resin you injected and what the volume of the original injection holes was. You could use this to calculate the volume of resin that has penetrated into the rock fractures at least.

► The ‘concretion-forming resin’ was injected by simple hand-operated equipment with minimal pressurisation (see text Line 9-10, p5). The cross-cut view of the core obtained by over-coring shows that the resin remained near the injection hole and did not extend beyond the over-cored core in the surrounding rock. This was confirmed by rock thin-sectioning. Furthermore, volume of injected resin in the hole is almost the same as the volume of the hole (difference is less than a few %). Thus the experimental design and the observations made on the over-cored rock together show that the resin did not extend radially very far from the resin injection hole and certainly not as far as 20 cm. An explanation has been added in Line 18-20, p6.

Which borehole did you over-core (on Figure 2) and where did you drill the hole that is 20cm away? It is not clear. I presume it is one of the core on SI Fig 3, but it doesn’t say.

► The locations of the resin injection holes has been added in Figure 2. Over-cores have obtained from around the No.5 and 6 resin injection holes. The No. 6 resin injection hole has the borehole 20 cm distant that is mentioned in the text. An explanation also has been added in captions of Figure 2 as well as SI Figure 3.

In the new Figure 5, where are images 5c, d, and e on Figure 5a? It is not clear where each sample is located. Figure 5f is really helpful, thank you. Where is the process described in 5f is located with respect to the injection hole? The resin itself is not marked on 5f. Is this a

cartoon of processes beyond the volume into which the resin has penetrated? Where the resin is present in the fractures, does it coat the fracture surfaces?

► The locations of Figure 5c, d and e are indicated in Figure 5b and show where high analysed Ca concentrations pick out EDZ flow-paths clogged by CaCO₃. Figure 5f is a cartoon to show the general image of calcite formation and the clogging characteristics of the fracture flow-paths remote from the resin injection hole, therefore Figure 5f has no resin shown (as noted above, the calcite is observed to have formed beyond the resin front). An explanation has been added to the caption of Figure 5.

In Figure 5d, is it resin that surrounds the CaCO₃ crystals, or is it void space in the fracture? It isn't labelled.

► The carton shows CaCO₃ crystals filling the void space of a flow-path in the rock matrix far from the resin injection hole. An explanation has been added in the caption of Figure 5.

Where are Figure 4c, d, e and f taken from? It is not clear whether these images are actually within the injection hole (now filled with resin) or whether they are located in the over-cored rock that surrounds it. For example, you show evidence in Figure 4 for CaCO₃ crystals within the resin, but it is not clear where the image was taken from. Is the resin within the injection hole or is it taken from a fracture in the over-cored rock volume?

► Figure 4c shows the flow-paths observed in core drilled 20 cm away from the resin injection hole No.6. The flow-path surface is covered by only calcite (without resin). Figures 4d and e are both SEM photos of calcite filling the same flow-path fracture as shown in Figure 4c. Figure 4f is the calcite formed inside the resin injected in the No.6 hole. An explanation has been added in the caption of Figure.4.

Response letter: "Calcite formation and cementation over time have been confirmed by laboratory experiments using concretion-forming resin, distilled water and glass beads...We think that the same sealing process as observed in the laboratory experiment has occurred in the in-situ test". Where is the evidence for this in the paper or in the references? The letter doesn't give any clues. I presume it is Figure 12 in the SI.

► This is from the Figure 12 in the SI.

What is the mass of calcite precipitated, with respect to the mass of resin in Fig 12 over the 3 month period. How does this compare to the Ca mass loss rate in the resin in the field trial? Is the field trial slower, or as expected?

► The purpose of the lab experiment detailed in the SI was to show the sequential calcite precipitation and occurrence of sealing between glass beads as a function of time, rather than to quantify the process by using the data obtained mass balanced calculation or numerical modeling.

Reviewer #2' comments:

The paper would be much stronger if they included some quantitative estimates of what they would expect for calcium precipitation and compared the field data they do have to expectations from their lab experiments – as suggested in my previous review. They are arguing this is not required and that the observations of Ca precipitation are enough.

I leave it up to the Editor whether you are happy to publish based on the field observations only, with no attempt at quantification, or not.

15th/Feb/2024

Dear Editor of *Communications Engineering*,

We have no comments for the Editor.

Answer to reviewers

Reviewer #2' comments:

The paper would be much stronger if they included some quantitative estimates of what they would expect for calcium precipitation and compared the field data they do have to expectations from their lab experiments – as suggested in my previous review. They are arguing this is not required and that the observations of Ca precipitation are enough.

Thank you for your suggestion to add quantitative estimates of our expectations for calcium precipitation. After discussions among my co-authors I have revised the paper to include quantitative estimates of the volume of calcite that could precipitate, and the distances over which this precipitation could occur. These estimates use the measured degree of Ca leaching from the resin and are based on a diffusion model and mass balance. We have also added a comparison between the quantitative estimates and the observations made in the in-situ experiment described in the paper.

These quantitative estimates, and arguments based on them, are presented in the main part of the text (Page 8 L29 ~ Page 10 L32). The conclusions have also been revised accordingly (Page 11 L31 ~ Page 12 L4).

Both the mass balance and diffusion calculations provide estimates of the rock volume in which fractures were sealing by calcite. These estimates are broadly consistent with the observations.

Also some related references and description of the quantities of Ca leached have been added to the text (Page 5 L9~11), and the explanation of SXAM has been expanded (Page 6 L37 ~ Page 7 L2). Figures 4 and 5 and the caption were also corrected.

Reviewers' comments:

Reviewer #4 (Remarks to the Author):

The paper presents clear results of an experiment executed in URL. It has great benefit of reporting also results from unplanned earthquake effects which adds to the significance of the results.

The paper shows an excellent example of how learning from natural systems can help to solve technical issues. This is a great merit for this work. This approach also allows using natural analogues to be assessed regarding the long-term behaviour of the grout, which is important especially in the field of nuclear waste disposal.

The conclusions should be interesting to the greater community of underground engineering solutions.

I have made some specific comments for further improving the manuscript in an attached file. Minor revision is required.

The main comment is related to the discussion of other sites and groundwater regimes. Currently, this discussion is limited and would benefit from more thorough discussion. However, the manuscript has relevant results that could only be reported in the context of the URL. However, some conclusions could be rephrased to reflect either the site specificity and mentioning similar sites, or if more references are added, the applicability to variable groundwater chemistries could be assessed. This will be left for the authors to decide. In any case, it will be interesting for further studies regarding applications in some slightly different chemical conditions (e.g. HCO_3 depleted areas common in many deep groundwaters). Also, some modelling work would be of interest in the future looking into the different groundwater chemistries elsewhere.

Overall, this paper increases the thinking in the field of rock engineering, that many processes have natural analogues, and it is beneficial to learn from them.

Dear Editor of *Communications Engineering*,

We have no comments for the Editor.

Answer to reviewer

Reviewer #4 (Remarks to the Author):

I have made some specific comments for further improving the manuscript in an attached file. Minor revision is required.

The main comment is related to the discussion of other sites and groundwater regimes. Currently, this discussion is limited and would benefit from more thorough discussion. However, the manuscript has relevant results that could only be reported in the context of the URL. However, some conclusions could be rephrased to reflect either the site specificness and mentioning similar sites, or if more references are added, the applicability to variable groundwater chemistries could be assessed. This will be left for the authors to decide. In any case, it will be interesting for further studies regarding applications in some slightly different chemical conditions (e.g. HCO₃⁻ depleted areas common in many deep groundwaters). Also, some modelling work would be of interest in the future looking into the different groundwater chemistries elsewhere.

► We think that the resin can be applied at locations where the groundwater differs widely in from that at the Horonobe test site, in terms of salinity and / or composition, because the resin itself can release the Ca²⁺ and HCO₃⁻ required to produce calcite. Therefore the method will work even it is used in a Ca²⁺ and HCO₃⁻ depleted environment. Evidence for this comes from our laboratory experiments which produced calcite on glass beads in the presence of distilled water. However, we agree with the reviewer that the application of the resin in different underground environments with different groundwater chemistries still requires further study. Text has been added to make these points, as suggested by the reviewer (p.9 L.1 - 5).

Reviewer #4' comments shown in the text:

<Page 1>

- 1) I am not sure if sealing all tunnels is needed for the long-term safety of nuclear waste disposal. The sentence could be reformulated that in "many underground applications some sealing of bedrock cavities may be required, e.g. bore-hole sealing...etc.... " For example at Olkiluoto, only some limitation is done for flow, but the aim is not to seal everything.
► We agree with the reviewer that "sealing", in the sense of emplacing effectively impermeable materials to prevent any fluid flow, is not necessarily required. Indeed, in this strict sense "sealing" is arguably impossible. Certainly, as mentioned by the reviewer, radioactive waste disposal facilities do not generally require such a high standard of sealing. However, invariably "sealing" will be required in the sense of emplacing materials to reduce fluid fluxes to some acceptable level. We have added a sentence at the end of p.1 (L37 - 39) to make this point clear.
- 2) demonstrating that self-healing sealing system can be achieved? It is not permanent if there is a local loss of sealing. I think the promising this here is the self-sealing / healing.
► We have modified the text to reflect the point made by the reviewer. Additionally, "permanent sealing" does not necessarily imply that no flow could ever occur, also described in the previous point. (p.1, L30).
- 3) rock mass?
► Corrected (p.1, L30).

- 4) rapidly recovered?
 - ▶ Corrected (p.1, L30).
- 5) often? or sometimes? I would not say it is typical.
 - ▶ Corrected to "often" (p.1, L35).
- 6) NEA and IAEA are not regulators per se. I would refer to these as "international guidance".
 - ▶ We have corrected as the reviewer suggested (p.1, L35).
- 7) potential sealing (not always done)
 - ▶ We have corrected as the reviewer suggested (p.2, L3).

<Page 2>

- 8) would be good to mention examples, at least OPC based and low-ph materials.
 - ▶ We have added examples of these materials as the reviewer suggested (p.2, L8).
- 9) these are rather old references. should perhaps use "e.g."
 - ▶ We have added "e.g." (p.2, L10)
- 10) only one ref?
 - ▶ We have added references (p.2, L11).

11) would be good to add these requirements here.

For example Posiva does not require this at all:

"The deposition tunnel plug consists of a concrete plug (L4-PLU-4).

The deposition tunnel concrete plug is only a supporting structure, not a barrier, and thus does not have long-term safety functions or performance targets, but due to its close proximity to the deposition tunnel backfill, its design requirements and specifications are included"

This is from: Posiva 2021. Safety Case for the Operating Licence Application - Design Basis (DB). POSIVA 2021-08. Eurajoki, Finland: Posiva Oy. Access via cms.posiva.fi

▶ We have not made any change to address this comment because the existing text adequately captures the point made, and because we wish to highlight the general applicability of our sealing technology. Our previous sentence states that cementitious seals *MAY* not be sufficiently durable *WHERE* long-term seal performance is required. The statement we have made is not inconsistent with the reviewer's comment. Our existing text allows for the possibility that long-term seal performance may not be required (as in Posiva's plugs), in which case seal durability would not be an issue. However, there are plenty of applications that do require long-term performance of cementitious seals. There are on-going debates about the effectiveness of cementitious seals in wells connected with CO₂ storage for example. Given the general nature of the statement we make and considering that it covers the possibility that not every application requires long-term durability, we think it is inappropriate to single out the example of Posiva's plug seal design for the Olkiluoto radioactive waste repository.

- 12) swells upon hydration and, high enough dry density provided, develops swelling pressure.
 - ▶ We have added the sentence suggest by the reviewer (p.2, L30 - 31).
- 13) the pressure does not seal apertures, just swelling. delete "pressure."
 - ▶ We have deleted "pressure" (p.2, L31).
- 14) or natural. does not have to be only EDZ derived.
 - ▶ We have corrected the text as the reviewer suggested (p.2, L32).
- 15) or better "certain types"
 - ▶ We have corrected the text as the reviewer suggested (p.2, L33).

- 16) more references is needed here. This has been studied in many URLs?
▶ We have added references (p.2, L35).
- 17) and also because the flow is wanted to be limited in general, also due to its potential effects on the EBS.
▶ We have added a statement to make clear that a purpose could be to reduce flow of water sufficiently to prevent detrimental effects on the engineered barriers (the sentence as the reviewer suggested p.2, L36 - 37).

<Page 3>

- 18) "have shown" would sound better
▶ We have corrected the text as the reviewer suggested (p.3, L13).
- 19) where is this shown?
▶ We have added the HP-URL and reference (p.3, L38).

<Page 4>

- 20) how about very saline groundwaters?
▶ There is no reason to suppose that the sealing technology would not work in saline water. There is abundant evidence that calcite concretions have formed naturally in the presence of seawater-derived pore water by a mechanism very similar to the one involved in calcite formation around the resin investigated in our study. Indeed, it was these natural observations that suggested the method to us in the first place. It is worthwhile noting that in Japan groundwaters more saline than seawater are quite spatially restricted; there are no evaporite deposit-derived brines. However, application of the sealing technology in the presence of brines is a topic for future investigation. We have added a statement to this effect (p.4 L. 20, 22 - 23).
- 21) ~10⁻⁶ to 10⁻⁵?
▶ We have corrected this (p.5, L4).
- 22) define what is BTV
▶ BTV has been defined (p.5, L7).
- 23) 5-10 or about 5-10 which would be ~5 to 10?
▶ We have corrected this (p.5, L9).

<Page 5>

- 24) clarify please. Can you add a reference to Japanese scale and explain how it differs from Richter scale?
▶ The earthquake on the Japanese intensity scale has been related to the internationally used Modified Mercalli Scale. The magnitude given is moment magnitude, Mw. The text has been modified to make these points (p 5, L27 - 28).
- 25) is this the same as optical borehole imaging?
▶ BTV is optical borehole imaging.
- 26) define earlier.
▶ Defined earlier (p.5, L41).
- 27) delete word "here"
▶ Deleted "here"
- 28) It would be helpful if the Figure showed Ca as well. Now only Si Al Fe Mn and K are shown, and their absence in the blue areas, but it would be clearer to show Ca. Add a reference to Figure 5 here where Ca is shown.
▶ We have added a reference to Figure 5 in the caption of Supplementary Figure 11.

<Page 6>

- 29) In Fig 12 the leaching does not seem to have stopped yet. This would imply even longer times of resin "activity"?
- ▶ The point made by the reviewer is exactly the one we make; at the end of the leaching experiment, leaching has not yet stopped. This is why the remaining Ca in the resin was analysed; the remaining quantity enables us to estimate how long leaching would have continued. We have modified the text to make this clearer (p.6, L. 12 - 14).
- 30) in the resin? Consider adding clarification.
- ▶ Clarification shown in the text (p.6 L.20).

<Page 7>

- 31) is this really diffusion?
- ▶ The hydraulic conductivity of the rock matrix is quite low, and there is insignificant driving head gradient. Distances over which calcite has formed are consistent with expectations based on the measured diffusivity of the formation, as shown by the calculations we present. For these reasons, we believe that diffusion is the dominant solute transport mechanism.
- 32) replace with "the"
- ▶ We have corrected this (p.7. L.11).
- 33) afterwards
- ▶ We have corrected this (p.7. L.14).
- 34) clarify what does this sentence relate to, or can it be left out?
- ▶ The sentence has been deleted.
- 35) or "was left to evaluate"?
- ▶ It was left to evaluate. We have corrected this (p.7. L.16).
- 36) is expected to proceed? I think this is still to be shown after longer monitoring.
- ▶ Yes, we think too. We have modified the statement as the reviewer suggested (p.7. L.35).
- 37) This "from these fractures" can be deleted for readability.
- ▶ We agree; it has been deleted.
- 38) or might have caused new transmissive fractures to open... perhaps clearer.
- ▶ We have corrected the text as the reviewer suggested (p.8. L.6 - 7).

<Page 8>

- 39) I would think it is also dependent on the groundwater chemistry? Was lab experiments done in distilled water? Discussion of overall effect of groundwater chemistry would be beneficial. Or at least, it should be pointed out as topic for further study?
- ▶ The lab experiment was done in distilled water. Even though the Ca and HCO₃ required come entirely from the resin, the experiment showed that calcite would nevertheless form. It follows that in the presence of any groundwater containing Ca and HCO₃, greater quantities of calcite would form than in the experiment. Therefore, in more saline water, the sealing technology is expected to be, if anything, more effective than in the experiment. However, we agree with the reviewer that groundwater chemistry is an important factor that requires further study. The necessity for further study has been indicated in the text (p.9, L1 - 6).

<Page 9>

- 40) These interactions should be defined.
- ▶ We have added explanations (p.9 L31).

<Page 10>

- 41) Hence
▶ We have corrected this (p.10 L32).

<Page 11>

- 42) what is the groundwater condition at the site?
▶ The groundwater is of sea-water origin. This has been mentioned in page 4, L33 - 35.
- 43) there are also other ones, such as silica based grouts, but maybe off-scope here.
▶ We know other types of grouting materials, but here we focus on calcium carbonate based material.
- 44) I think this statement is too vague. The rocks are not defined, or references given for groundwater conditions. In addition, reference 54 is a poster? I could only access the abstract, but there was no such conclusions. I think this should be conclusion of this paper.
▶ We agree and this part has been moved to the conclusion (see below). Also reference 54 was deleted (p.12 L 29 - 32).
- 45) This is a very important point. This is excellent use of analogues not just to address long-term stability but also as a source of ideas for engineering.
▶ This part has also been moved to the conclusion (p.12 L 29 - 32).

<Page 12>

- 46) This should not come first time here in conclusions. It should be moved to discussion. No new references should be given in conclusions.
▶ This part has moved to discussion (p.11 L 24 - 27).
- 47) In addition bicarbonate is commonly absent in deep groundwaters.
▶ As noted above (and pointed out also in the paper), it is not necessary for the groundwater to contain bicarbonate in order for the resin-based sealing technology to work, though certainly it would be beneficial if bicarbonate is present. We agree with the reviewer that in many places around the world, there is little bicarbonate in deep groundwaters at depths that are relevant to many activities where sealing might be required, such as deep radioactive waste disposal (typically proposed at depths of around 500 m). However, it is not strictly true that at such depths bicarbonate is commonly *absent*. Even at Olikiluoto, the site mentioned by the reviewer, deep saline waters have in the order of 10^{-4} mol / L bicarbonate (e.g. Idiart et al., 2013, Posiva Report POSIVA 2013-05). In Japan, groundwaters at depths of 500 m typically contain significant concentrations of bicarbonate. For example, based on a countrywide statistical analysis of groundwater data Yui et al. (2004, Geochemical Journal, v 38) proposed reference compositions for saline Japanese groundwaters at depths >500 m having in the order of 10^{-2} mol/L bicarbonate.